# Research Progress and Challenges in the Treatment of Central Nervous System Metastasis of Non-Small Cell Lung Cancer

**DOI:** 10.3390/cells10102620

**Published:** 2021-10-01

**Authors:** Bin Wang, Hanfei Guo, Haiyang Xu, Hongquan Yu, Yong Chen, Gang Zhao

**Affiliations:** 1Department of Neurosurgery, The First Hospital of Jilin University, Changchun 130021, China; binwang19@mails.jlu.edu.cn (B.W.); xuhaiy@jlu.edu.cn (H.X.); yhq@jlu.edu.cn (H.Y.); 2Cancer Center, The First Hospital of Jilin University, Changchun 130021, China; guohf7011@mails.jlu.edu.cn

**Keywords:** central nervous system metastasis, non-small cell lung cancer, brain metastasis, leptomeningeal metastasis, radiotherapy, chemotherapy, targeted therapy, immunotherapy

## Abstract

Non-small cell lung cancer (NSCLC) is one of the most common malignant tumors and has high morbidity and mortality rates. Central nervous system (CNS) metastasis is one of the most frequent complications in patients with NSCLC and seriously affects the quality of life (QOL) and overall survival (OS) of patients, with a median OS of untreated patients of only 1–3 months. There are various treatment methods for NSCLC CNS metastasis, including surgery, chemotherapy, radiotherapy, targeted therapy, and immunotherapy, which do not meet the requirements of patients in terms of improving OS and QOL. There are still many problems in the treatment of NSCLC CNS metastasis that need to be solved urgently. This review summarizes the research progress in the treatment of NSCLC CNS metastasis to provide a reference for clinical practice.

## 1. Introduction

Lung cancer ranks first in terms of morbidity and mortality among all tumors worldwide, and non-small cell lung cancer (NSCLC) is the most common type of lung cancer [1]. The central nervous system (CNS) is a common clinical site for metastasis of NSCLC, which seriously affects the prognosis and quality of life (QOL) of patients. The incidence of CNS metastasis in patients with NSCLC at initial diagnosis is approximately 10% [2,3], and approximately 30% of patients with NSCLC develop CNS metastasis during the course of their disease [4,5]. NSCLC includes adenocarcinoma, squamous cell carcinoma, and large cell carcinoma, and the risk of CNS metastasis for each subtype is 11%, 6%, and 12%, respectively [3]. CNS metastases in NSCLC include brain metastasis (BM) and leptomeningeal metastasis (LM). BM most often occurs in the cerebral hemispheres, cerebellum, and brainstem [2]. On the other hand, LM refers to the spread of malignant tumor cells through the cerebrospinal fluid (CSF) to the leptomeninges (pia and arachnoid mater), which is a rare event with an incidence of only 3–5% in patients with NSCLC. The prognosis of patients with NSCLC LM metastasis is poor, with overall survival (OS) of 3 months with contemporary treatment and less than 11 months with novel therapies [6].

There are various risk factors for CNS metastasis in patients with NSCLC, including age, tumor type, histological grade, number of positive lymph nodes, and driver mutations [7,8]. In particular, the incidence of CNS metastasis is significantly higher in patients with epidermal growth factor receptor (EGFR) mutation, anaplastic lymphoma kinase (ALK) rearrangement, or human epidermal growth factor receptor-2 (HER-2) mutations [8,9,10,11,12].

The prognosis of patients with NSCLC CNS metastasis is poor. The median OS is approximately 1–3 months for untreated patients [13] and 7 months for treated patients [14,15]. Available treatment options for NSCLC CNS metastasis include surgery, radiotherapy, chemotherapy, immunotherapy, and targeted therapy (Figure 1). Surgical resection acts as a rapid steroid taper and allows for the relief of neurological symptoms, such as intracranial hypertension, neurological deficits, and seizures. Patients with NSCLC CNS metastasis treated with whole-brain radiotherapy (WBRT) alone generally have a poor prognosis with a median survival of less than 6 months [16]. Stereotactic radiosurgery (SRS) is a less neurotoxic alternative to WBRT with no difference in OS [17]. The role of systemic chemotherapy in the treatment of BMs is debatable, with the response rates (RRs) ranging from 15 to 30% (OS 6–8 months) [18,19]. The life span of patients with NSCLC CNS metastasis is significantly increased by the clinical application of targeted therapy and immunotherapy. Patients with NSCLC CNS metastasis harboring EGFR mutations have a great response to EGFR tyrosine kinase inhibitor (TKI) treatment with RRs of 60–80% (OS 15–20 months) [20,21]. Similarly, patients with ALK-rearranged NSCLC CNS metastasis have a dramatic response to ALK-TKI treatment with RRs of 36–72% (progression-free survival [PFS] 5.7–13.2 months) [22]. Immune checkpoint inhibitors (ICIs) have become the standard of care in patients with NSCLC CNS metastasis with a 5-year OS ranging from 15% to 23% [23].

The progressive deterioration of neurological and cognitive functions has a negative effect on the QOL of patients [24]. Progress in screening high-risk patients and the development of new therapies may improve patient prognosis. Magnetic resonance imaging (MRI) is widely used as a gold standard diagnostic and monitoring tool for NSCLC CNS metastasis. Choosing an appropriate treatment plan for patients with NSCLC CNS metastasis is a current clinical problem that needs to be solved urgently. This article reviews the treatment progress and prognostic factors associated with NSCLC CNS metastasis.

## 2. Local Treatment

Current local treatments for NSCLC CNS metastasis include surgery, WBRT, SRS, and stereotactic radiotherapy (SRT).

### 2.1. Surgery

Surgical removal of intracranial metastasis can quickly alleviate the neurological symptoms caused by tumor-related compression and obtain clear pathological evidence. The indications for NSCLC CNS metastasis-targeting surgery include 1–3 BMs, BM lesions with a diameter more than 3 cm, superficial tumor location, tumors located in non-functional areas, large metastasis in the cerebellum (diameter of >2 cm), and patients who cannot accept or have contraindications for corticosteroid treatment [13,25]. When there is non-obstructive hydrocephalus, high intracranial pressure symptoms (such as vomiting, papilledema, neck stiffness, and severe headache), or obvious ventricular dilatation that cannot be relieved by dehydrating agents, surgical intervention should be performed to relieve the CNS metastasis crisis [26,27]. Resection of metastatic brain lesions provides immediate amelioration of mass effect and neurological deficits and avoids the requirement of long-term steroid use, which in turn allows the early initiation of ICIs [28,29,30]. Advances in neurosurgical technologies such as neuronavigation, intraoperative ultrasound, fluorescence-guided surgery, and intraoperative neuromonitoring lead to safer surgery and reduce the risk of morbidity and mortality with total resection [2]. WBRT and SRS are effective treatment methods following surgery. SRS can offer a similar control rate of tumors as WBRT, with fewer side effects which make SRS a better choice [31].

### 2.2. Whole Brain Radiotherapy

Indications for WBRT in NSCLC CNS metastasis are as follows: three or more BMs and BM lesions less than 3 cm. WBRT can also be used as an adjuvant treatment after surgery or SRS. The total remission rate of WBRT treatment alone can reach 60%, which can prolong the median OS by 4–6 months, and the most common WBRT regimen uses 10 fractions of 3 Gy over 2 weeks (30 Gy) [32]. However, WBRT has greater side effects on the nervous system [33]. The Quality of Life after Treatment for Brain Metastases (QUARTZ) trial is a randomized phase III trial comparing best supportive care (BSC) plus WBRT versus BSC alone for patients with NSCLC CNS metastasis. The QUARTZ trial revealed that there is no detriment to QOL and OS for patients allocated to BSC alone among patients with NSCLC with unfavorable prognostic factors [34]. The use of drugs such as memantine [35] and donepezil [36] is expected to improve the neurocognitive dysfunction caused by WBRT, and related clinical studies (NCT02360215) are ongoing. Compared with SRS/SRT alone, SRS/SRT combined with WBRT can increase the control rate of intracranial lesions and incidence of neurocognitive impairment, although there was no difference in OS [37]. It is important to note that patients with NSCLC with actionable oncogenic driver alterations such as EGFR or ALK and asymptomatic or oligosymptomatic BM should be treated by upfront systemic targeted therapy rather than radiation therapy [38,39]. Therefore, the position of WBRT in the treatment of NSCLC CNS metasctasis is gradually being replaced by new therapies.

### 2.3. Stereotactic Radiosurgery and Stereotactic Radiotherapy

Both SRS and SRT are radiotherapy techniques that use stereotactic technology. These are accurate, safe, and rapid techniques that deliver high doses to target sites and low doses to normal tissues. In the study of Paul et al., the SRS dose is 18–22 Gy in SRS/SRT combined with WBRT and 20–24 Gy for SRS alone, and SRS alone resulted in less cognitive deterioration at 3 months [37]. For patients with oligometastatic disease, SRS/SRT can achieve similar prognostic results and a higher local control rate compared with surgery [40]. In the study of Paul et al., the postoperative SRS (12–20 Gy single fraction with the dose determined by surgical cavity volume) resulted in less cognitive deterioration and no difference in OS compared with WBRT for resected metastatic brain disease [17]. In the past, WBRT was the first choice for patients with multiple BMs; however, the JLGK0901 study showed that the OS of patients with 5–10 BMs following SRS treatment was 10.8 months, which was not inferior to patients with 2–4 metastases (hazard ratio (HR) 0.97, 95% confidence interval [CI] 0.81–1.18 (less than non-inferiority margin), *p* = 0.78; *p*_non-inferiority_ < 0.0001) [41]. The cumulative incidence of complications in the two groups was tracked for the following 2 years, and complications did not increase during this period, proving the efficacy and safety of treatment [42]. In a phase III randomized controlled trial NCT01592968 with 4–15 non-melanoma BMs, local control was 100% for the SRS group at 4 months and 95.5% for the WBRT group (*p* = 0.53) without a significant difference in median OS (*p* = 0.45). Moreover, the neurocognitive prognosis at 4 months improved in the SRS group [43]. Several ongoing trials will evaluate whether SRS alone can preserve the neurocognitive function with no reduction in local control and OS compared with WBRT for patients with up to 15–20 brain lesions (NCT01592968, NCT03075072, NCT03550391, NCT03775330). Therefore, SRS can be used for patients with multiple BMs.

Owing to the excellent tumor control and minor neurocognitive side effects, SRS/SRT has gradually become one of the main treatments for NSCLC CNS metastasis in recent years. SRT treatment is preferred for patients with NSCLC with stable primary lung tumor control, good performance status, 1–4 brain parenchymal metastases, and no metastasis to other parts of the body [44].

### 2.4. Exploration of New Local Treatment Methods

A variety of reformed radiotherapy methods have been proposed to improve the control rate of BMs and to protect the neurocognitive function of patients. Some examples are intensity-modulated radiotherapy (IMRT), volumetric-modulated arc therapy (VMAT), simultaneous modulated accelerated radiation therapy for the brain (SMART-Brain), and hippocampal-avoidance WBRT (HA-WBRT) [45]. Compared with WBRT, HA-WBRT reduces the radiation dose to the neural stem cell compartment in the hippocampus by 80% and limits the negative effects on neurocognitive function without affecting the patient’s OS and low-dose area recurrence rate. HA-WBRT also effectively improves the patient’s short- and long-term QOL [46]. In addition, the combination of VMAT and an automated treatment planning system can further reduce the radiation dose in the hippocampus, improve dose homogeneity, and decrease unnecessary hot spots in the healthy brain [47]. SMART-Brain is a brain radiotherapy method based on IMRT that implements increased irradiation of BMs and protection of key functional areas. SMART-Brain protects the hippocampus (≤10 Gy) and inner ears (≤15 Gy) under the premise of WBRT (30 Gy/10F/2 weeks) and brain metastatic lesions high-dose radiotherapy (40–50 Gy/10F/2 weeks) [48]. Related multicenter randomized controlled studies (CRTOG1702/1703) are ongoing.

## 3. Chemotherapy

Cytotoxic therapy has a controlling effect on NSCLC CNS metastasis without driver mutations or in patients who do not meet other therapeutic indications. Platinum combined with pemetrexed can confer survival benefits to patients with NSCLC CNS metastasis. The study of Barlesi et al. suggests that the objective response rate (ORR) to cisplatin combined with pemetrexed for intracranial lesions can reach 41.9% [49]. In another phase II clinical study, patients with NSCLC BMs who received high-dose pemetrexed combined with cisplatin maintenance therapy after WBRT had an ORR of 68.8%, while the median PFS and median OS were 13.6 and 19.1 months, respectively [50]. Temozolomide (TMZ) is an oral alkylating agent that can penetrate the blood–brain barrier (BBB) and has a good effect in controlling CNS metastasis in NSCLC. TMZ alone or combined with other chemotherapeutic drugs together with sequential WBRT or simultaneous WBRT can improve the ORR of patients with NSCLC CNS metastasis [51].

## 4. Targeted Therapy

NSCLC is a highly heterogeneous cancer with several molecular subtypes related to specific driver genes, which have different prognoses and treatment responses [52]. TKIs, such as EGFR-TKIs and ALK-TKIs, that target NSCLC driver mutations have greatly improved the prognosis of patients with NSCLC CNS metastasis with the corresponding gene mutations. Thus, TKIs are recommended as the first choice for the treatment of NSCLC CNS metastasis with driver mutations, such as those affecting EGFR or ALK [52,53]. In addition, the concentration of TKIs in the blood and CSF is an important indicator in predicting treatment efficacy. The vascular endothelial growth factor (VEGF) antagonist, bevacizumab, combined with chemotherapy, also shows positive clinical effects in patients with NSCLC CNS metastasis without driver mutations [54,55,56].

### 4.1. Targeted Therapy with EGFR Tyrosine Kinase Inhibitors

*EGFR* mutations are the most common type of mutation in patients with metastatic NSCLC, accounting for approximately 50% of cases in Asia [57]. The presence of *EGFR* mutations is correlated with an increase in OS [58]. In addition, *EGFR* mutations are also associated with an increase in the incidence of NSCLC BMs compared with *EGFR* wild-type group (odds ratio (OR) = 2.01; 95% CI, 1.56–2.59; *p* = 0.000) [8]. NSCLC CNS metastases with *EGFR* mutations are characterized by multiple scattered small metastases with less peritumoral edema [59].

First-generation EGFR-TKIs (gefitinib, erlotinib, and icotinib) and second-generation EGFR-TKIs (afatinib and dacomitinib) have poor BBB permeability and provide a higher ORR of approximately 60% of intracranial lesions in NSCLC CNS metastasis compared with that of WBRT with or without chemotherapy (ORR < 40%) [60,61,62,63,64]. Studies on EGFR-TKIs in patients with NSCLC CNS metastasis show that pulsed high-dose erlotinib or gefitinib can increase the drug concentration in the CSF [65,66] and effectively induce tumor cell apoptosis [67]. Patients with LM may also benefit from these drugs [68,69], although treatment-related adverse events (AEs) lead to a high rate of drug withdrawal [65]. The pulsed high-dose erlotinib dose-escalation phase I trial was terminated early because of its limited efficacy [70].

The third-generation EGFR-TKI, osimertinib, is a mutant-selective EGFR inhibitor that can irreversibly inhibit NSCLC even in the presence of EGFR-sensitizing mutations and T790M resistance mutations. Osimertinib has a better BBB permeability and thus has a higher concentration in the CSF than the first two generations of EGFR-TKIs [71]. The FLAURA trial showed that osimertinib is more effective than the current standard first-line treatment (erlotinib or gefitinib). The data also revealed that the PFS in the osimertinib treatment group was 18.9 months, which was significantly longer than that in the control group (10.2 months), and the incidence of serious AEs was 10.6% lower [72,73,74]. The median OS in the osimertinib group was 38.6 months, which was significantly higher than that in the standard treatment group (31.8 months). In addition, 28% of patients in the osimertinib group continued to receive the trial regimen after 3 years of treatment, which was significantly higher than 9% in the standard treatment group [75]. Osimertinib also significantly improved the prognosis of patients with NSCLC. A study of 351 patients with NSCLC LM showed that patients treated with osimertinib had a median OS of 17.0 months (*n* =110), which was approximately three times higher than that of patients who did not receive osimertinib (*n* =241) (17.0 months vs. 5.5 months; HR = 0.38; 95% CI, 0.28–0.47; *p* < 0.001) [76]. The same study found that the disease control rate (DCR) reached 91%, among which less than 30% of the patients received osimertinib as first-line therapy [76]. Currently, there is a lack of prospective data on LM patients with first-line osimertinib treatment, and this needs to be addressed in future studies. Nevertheless, osimertinib has been approved as the first-line treatment for NSCLC with EGFR mutations, breaking the sequential pattern of NSCLCs.

Furmonertinib (Alflutinib, AST2818), a newly developed third-generation EGFR-TKI, treats NSCLC CNS metastasis with the *EGFR* T790M mutation [77]. A study of 220 patients with NSCLC with *EGFR* T790M mutations showed that patients treated with furmonertinib had an ORR of 74% [78]. In the study of Yuankai et al., 130 patients (14 in dose escalation and 116 in dose expansion) received furmonertinib orally. In the dose-expansion group, the overall ORR was 76.7% (89 of 116), and the ORR of CNS metastasis was 70.6% (12 of 17) [79]. The ORR of 80 mg furmonertinib treatment for NSCLC CNS metastasis reached 60.0%, the ORR of 160 mg furmonertinib treatment reached 84.6%, and the DCR was 100% [79].

### 4.2. Targeted Therapy with ALK-TKI

Although NSCLC with ALK rearrangement accounts for only a small proportion (4–8%) of all patients with NSCLC, this is an important subgroup with different epidemiological and biological characteristics [80]. Patients with NSCLC with ALK rearrangement are younger and usually have no or light smoking history [9]. ALK rearrangement is associated with an increase in the incidence of BMs in patients with NSCLC, and the cumulative incidence of post-diagnosis BMs at 2 and 3 years is 45.5% and 58.4%, respectively [81].

The first-generation ALK-TKI, crizotinib, was the first ALK inhibitor approved for the treatment of patients with metastatic ALK-positive NSCLC, which can induce ALK, c-MET, and ROS-1 fusion protein inhibition [39]. Patients develop crizotinib resistance due to the presence of secondary mutations in the ALK kinase domain and the drug’s inability to cross the BBB [82]. The most common treatment complication is intracranial progression [83]. A retrospective study showed that 20% of patients with NSCLC without BMs at baseline developed BMs during crizotinib treatment, and 72% of patients with NSCLC with BMs at baseline developed secondary BMs during crizotinib treatment after controlling for intracranial lesions [84]. These issues with crizotinib treatment necessitate research on second-generation ALK-TKIs, which could be effective alternatives.

Second-generation ALK-TKIs (alectinib, brigatinib, and ceritinib) have better BBB permeability, allowing them to control brain lesions well and to provide a single-drug treatment option [85,86]. If the maximum diameter of the brain lesion is reduced by less than 30% after 1–3 months of second-generation ALK-TKI treatment, radiotherapy should be added [27]. A phase III ALUR study showed that patients with ALK-positive NSCLC BMs treated with alectinib had a significantly higher ORR than patients who underwent chemotherapy (54.2% vs. 0, *p* < 0.001) [87]. Regardless of the baseline BM or prior radiotherapy, alectinib is more effective than crizotinib [83,86,87,88,89]. The J-ALEX study showed that alectinib can significantly reduce the risk ratio of intracranial metastasis progression compared with crizotinib (HR=0.51 for patients with BM at baseline; 95% CI, 0.16–1.64; *p* = 0.2502; and HR = 0.19 for patients without BM at baseline; 95% CI, 0.07–0.53; *p* = 0.0004) and 1-year cumulative incidence rate of CNS metastasis (5.9% vs. 16.8%) [90]. A phase I/II randomized clinical study of Gettinger et al. and phase II ALTA study Kim et al. showed that brigatinib can produce significant intracranial ORR in patients with ALK-positive NSCLC with intracranial progression or relapse after crizotinib treatment (I/II stage: 53%, ALTA arm A: 46%, ALTA arm B: 67%) and improved intracranial PFS (I/II stage: 14.6 months, ALTA arm A: 15.6 months, ALTA arm B: 18.4 months) [91]. Ceritinib also provided significant clinical benefits in patients with ALK-positive NSCLC after the failure of crizotinib treatment [92]. The ASCEND-2 study included 140 patients with ALK-positive NSCLC who progressed during crizotinib treatment, and 71.4% of patients (100/140) had BMs. The ORR of patients receiving ceritinib for BMs in the ASCEND-2 group was 33%, and the median PFS was 5.4 months [93]. The ASCEND-4 study showed that for patients with BMs at baseline, the intracranial ORR was 72.7% in the ceritinib group and 27.3% in the chemotherapy group, and the median PFS was 10.7 months and 6.6 months, respectively [94].

The third-generation ALK-TKI, lorlatinib, is a small-molecule dual-target inhibitor of ALK and ROS-1 that competes with ATP and has both high efficiency and selectivity. It is designed to pass the BBB and to overcome ALK-TKI resistance due to the G1202R mutation [95], and it shows better CNS efficacy in patients with NSCLC [96]. The results of a phase II clinical study of Benjamin et al. showed that the intracranial ORR of ALK-positive patients with NSCLC treated with lorlatinib was 66.7% in treatment-naive patients and 63% in patients with at least one prior ALK-TKI treatment [97].

### 4.3. Other Targeted Therapies

Bevacizumab is a recombinant humanized monoclonal antibody that can selectively bind VEGF and reduce the formation of tumor blood vessels, thereby inhibiting tumor growth. The combination of atezolizumab and bevacizumab with chemotherapy is a therapeutic option for patients with NSCLC CNS metastasis without driver mutations [53,98,99]. The results of several retrospective clinical studies have shown that the efficacy of bevacizumab is similar for intracranial and extracranial lesions, and the incidence of brain metastasis in bevacizumab plus chemotherapy is 17% less than that in chemotherapy alone [100]. A retrospective study of 776 patients with NSCLC BMs showed that the efficacy of bevacizumab combined with chemotherapy was better than that of chemotherapy alone, TKIs alone, or supportive treatment. The same study found that the median PFS and median OS of patients treated with bevacizumab plus chemotherapy were 8.5 months and 10.5 months, respectively, which was greater than those with the other three therapies with or without EGFR mutations (*p* < 0.01) [101]. There are many other studies on bevacizumab in progress (NCT04345146, NCT02681549, NCT02971501, and NCT04213170).

Other NSCLC-related driver mutations act as potential therapeutic targets for NSCLC and help in controlling BM. These include *ROS-1*, *HER-2*, *RET* proto-oncogene, mesenchymal-epithelial transition factor receptor tyrosine kinase gene (*MET*), v-Raf murine sarcoma viral oncogene homologue B1 (*BRAF*), and tyrosine kinase receptor B (*TrkB*) [102,103,104]. Experts consider the prevention, delay, and treatment of NSCLC CNS metastasis as a focus for future research, in addition to ongoing related studies.

## 5. Immunotherapy

With the development of ICIs, ICI monotherapy or in combination with chemotherapy has become the first-line treatment strategy for patients with metastatic NSCLC. ICIs activate T cells to cross the BBB [105] and thus have a certain effect against CNS metastases. A retrospective study of 255 patients with NSCLC BMs showed that the intracranial ORR was 27.3% after first-line treatment with ICIs [106]. Immunotherapy has gradually become an important treatment for CNS metastasis in NSCLC without driver gene mutations (Table 1).

### 5.1. Treatment Progress of ICI Monotherapy in NSCLC CNS Metastasis

Pembrolizumab monotherapy as second-line treatment has an intracranial ORR of approximately 20–30% in patients with NSCLC BMs [133]. Pembrolizumab has similar efficacy in intracranial and extracranial lesions in patients with NSCLC BMs. A phase II clinical study of 34 patients with NSCLC BMs showed that the median OS was 7.7 months, and both the intracranial and overall ORR were 33% [134].

Retrospective analysis shows that the survival benefit of patients with advanced NSCLC treated with nivolumab has little to do with the presence of BMs [114]. The CheckMate-017 study, CheckMate-057 study, and CheckMate-063 study are the three main clinical studies of nivolumab monotherapy as a second-line treatment of advanced NSCLC. A meta-analysis of 88 patients in the BM subgroup showed that patients with NSCLC BMs treated with nivolumab had a better OS (8.4 months vs. 6.2 months), a delayed occurrence of new intracranial lesions, and a lower incidence of new BMs at 6 months (13% vs. 17%) [135]. In the phase I multicohort CheckMate-012 study, the cohort included 12 newly treated patients with asymptomatic NSCLC with BMs. After treatment with nivolumab alone, the ORR was 16.7%, the DCR was 16.7%, the median OS was 8.0 months, and the median PFS was 1.6 months [135]. A retrospective study of the nivolumab expanded access program included patients with advanced lung squamous cell carcinoma (*n* = 371) and non-squamous NSCLC (*n* = 1588). The results showed that nivolumab has similar benefits in advanced lung squamous cell carcinoma and non-squamous cell NSCLC, with a total DCR of 49% and 40% and CNS ORR of 19% and 17%, respectively [136].

The OAK study results showed that compared with docetaxel, atezolizumab treatment of NSCLC BMs led to better median OS (16.0 months vs. 11.9 months, HR = 0.74, *p* = 0.1633) and fewer reports of treatment-related AEs, serious AEs, and treatment-related neurological AEs. Atezolizumab also had demonstrated preventive effects against new BMs (median time to new brain metastases: <9.5 months, HR = 0.38, *p* = 0.0239) [137]. In the phase II clinical FIR study, the ORR of 13 asymptomatic patients with NSCLC BMs treated with atezolizumab was 23%, and the median OS and median PFS were 6.8 months and 4.3 months, respectively [120].

Monotherapy can directly determine the efficacy of a drug. These small sample sizes and prospective studies suggest that the short-term efficacy of ICIs in the treatment of intracranial lesions in patients with NSCLC BM is similar to that of extracranial lesions; however, the PFS and OS are shorter, which may be due to the small sample bias. Additionally, patients with symptomatic BMs are often excluded from clinical studies. The efficacy of ICI monotherapy for NSCLC BMs needs to be further confirmed in large-sample prospective studies.

### 5.2. Treatment Progress of ICI Monotherapy Combined with Chemotherapy/Radiotherapy for NSCLC CNS Metastasis

A retrospective study showed that pembrolizumab plus chemotherapy compared with chemotherapy alone can increase the ORR of patients with BMs (80% vs. 58.3%, *p* = 0.75) and reduce the progression rate of BMs (33.3% vs. 91.7%, *p* = 0.009) [138]. The KEYNOTE-189 study, which included 108 patients with EGFR/ALK-negative non-squamous NSCLC BMs, reported that pembrolizumab combined with platinum and pemetrexed significantly improved the OS compared with chemotherapy alone (19.2 months vs. 7.5 months) [139].

The 2019 ASCO meeting retrospectively analyzed the data of 13,998 patients with NSCLC from the National Cancer Database, and it showed that patients with NSCLC BMs treated with immunotherapy plus intracranial radiotherapy had a longer median OS than patients treated with intracranial radiotherapy alone (13.1 months vs. 9.7 months) [140]. The results of the retrospective analysis of the American Hopkins Hospital on SRS/SRT treatment of tumor patients with BMs also suggested that immunotherapy combined with simultaneous SRS/SRT can improve OS and reduce the incidence of new BMs [141]. The time window for radiotherapy combined with immunotherapy is worth exploring. A retrospective study by the Moffitt Cancer Center in the United States showed that immunotherapy combined with radiotherapy, especially receiving SRS before or simultaneously with immunotherapy, can significantly improve the intracranial control rate compared with radiotherapy alone (57% vs. 0%) [142]. In terms of safety, a retrospective study of 54 patients with NSCLC BMs showed that there was no significant difference in the incidence of radiation necrosis or intratumoral hemorrhage between the immunotherapy plus SRS (37 cases) and SRS groups (17 cases) (5.9% vs. 2.9%, *p* = 0.99). Additionally, no significant difference was found in the incidence of peritumoral edema (11.1% vs. 21.7%, *p* = 0.162) [143]. However, another retrospective study involving 294 patients with NSCLC BMs showed that immunotherapy combined with radiotherapy increased the risk of symptomatic radiation necrosis (20% vs. 6.7%, *p* = 0.004), which was found to be related to immunotherapy [144]. The treatment directions of patients with BMs have diversified. Immunotherapy plus chemotherapy or radiotherapy has shown good clinical benefits. However, there is a need to explore the patients, timing, and AEs associated with combination therapy.

## 6. Discussion

### 6.1. Choice of Clinical Treatment Model for NSCLC CNS Metastasis with Driver Mutations

Owing to their small molecular weight, good lipid-to-water ratio, and strong BBB permeability, TKIs have greatly contributed to the progress of treatment of patients with EGFR-positive NSCLC CNS metastasis; however, driver mutations often mean an increase in the incidence of BMs [8,9]. The ability of different TKIs to pass through the BBB varies (Table 2). Most TKIs with better BBB permeability have good control of brain lesions in patients with NSCLC and have the effect of delaying the occurrence of BMs even with monotherapy [85,86]. If the maximum diameter of the brain lesion is reduced by less than 30% after 1–3 months of ALK-TKI treatment, radiotherapy should be added [27]. Crizotinib has low BBB permeability [82], and the probability of BMs occurring or progressing after crizotinib treatment in patients with ALK-positive NSCLC is higher [83,84]. Therefore, simultaneous radiotherapy is recommended when crizotinib is used for treatment.

The clinical treatment strategy for asymptomatic patients with BM is also controversial, especially regarding the choice of radiotherapy intervention. Some early studies have shown that radiotherapy does not improve the local control rate, OS, or QOL of patients with NSCLC. Radiotherapy-related AEs may also increase patient distress. Therefore, clinicians often use symptoms and progression as indications and standards for local treatment (SRT/SRS) intervention. TKIs should be used for patients with asymptomatic BMs, and radiotherapy should be performed after symptoms appear or progress. However, at the same time, studies have shown that TKI resistance may lead to the development of radio-resistance, thereby reducing the efficacy of radiotherapy for BMs [156]. In addition to increasing the local control rate and alleviating local symptoms, local treatment can increase the depth of systemic treatment through its remote effect and also provide long-term survival benefits. Therefore, from the perspective of radiotherapy, early treatment for BMs is recommended to kill brain lesions as much as possible regardless of whether symptoms appear, increase the depth of treatment, and prolong the survival of patients. Studies have shown that if patients with NSCLC BMs treated with osimertinib have received brain radiotherapy in the past 6 months, the effective rate tends to be higher than that of patients without radiotherapy (64% vs. 34%) [157].

Several studies have shown that there are differences in the efficacy of radiotherapy combined with TKIs, which may be related to the small sample size of these studies. NSCLC CNS metastasis with driver mutations also needs further research and the accumulation of more clinical data to obtain a treatment model that is more appropriate for the disease.

### 6.2. Prognostic Factors of NSCLC CNS Metastasis

Improvements in the diagnosis and treatment methods have increased the detection rate of CNS metastases and prolonged survival time; however, the overall prognosis of patients with CNS metastases remains poor [158]. At present, the results of research on the prognostic markers of NSCLC CNS metastasis are quite different and controversial, and there is no unified conclusion.

The relationship between patient age and the prognosis of NSCLC CNS metastasis is unclear. A retrospective study of 491 patients with NSCLC BMs showed that the survival rate of young patients was higher than that of elderly patients [159]. However, another study of 105 patients with NSCLC BMs found no significant correlation between patient age and survival prognosis [160]. The control of primary lung lesions can significantly improve the 12- and 18-month survival rates of patients with NSCLC BMs [161]. However, several other studies have shown no significant correlation between the control of the primary tumor and survival prognosis [162]. This may be because there is no uniform standard for the definition of primary disease control. The evaluation of the relationship between the control of primary lung lesions and survival prognosis of patients with NSCLC CNS metastasis has led to conflicting conclusions in previous clinical studies, which need to be further confirmed in large-scale studies [163]. Patients with multiple BMs are more likely to have acute symptoms and a shorter survival period than patients with single metastasis. However, recent studies have shown that the average survival of patients with multiple BMs is not statistically different from that of patients with single metastasis [164].

*EGFR* mutations and ALK rearrangements are independent prognostic factors for NSCLC BM (HR = 0.5) [158]. Although targeted therapy and immunotherapy can benefit some patients with NSCLC CNS metastasis, it is still not possible to accurately select the ideal population. Therefore, there is an urgent need to find accurate biomarkers for the treatment of NSCLC CNS metastasis more precisely, individually, and optimally.

## 7. Conclusions

With the continuous emergence of new therapies, the systemic treatment of NSCLC CNS metastasis has undergone revolutionary changes, significantly improving the OS and QOL of patients. Owing to the development of gene sequencing technology, the treatment of lung cancer has gradually evolved from the macro to the micro scale, entering the era of precision treatment. Local treatments, such as radiotherapy and surgery, are the basis of the treatment of CNS metastases and are also the first choice for the treatment of BM crises. EGFR-TKI and ALK-TKI show good intracranial control rates and prolong survival in NSCLC CNS metastasis with corresponding driver gene mutations. The BBB penetration and CSF concentration of these TKIs are the main factors that determine efficacy. Targeted therapy has a quick but short-term effect, resulting in inevitable drug resistance. Anti-programed death-1 or anti-cytotoxic T-lymphocyte antigen 4 immunotherapy can achieve control of intracranial and extracranial disease with a long-lasting response; however, it often has a delayed onset. In clinical practice, a more personalized treatment plan needs to be selected according to the patient’s pathological type, general condition, graded prognostic assessment score, and other important factors. At the same time, a corresponding clinical research design is needed to determine the best treatment plan for patients with NSCLC CNS metastasis.

Despite the high incidence of CNS metastases, patients with untreated and/or symptomatic CNS metastases are often excluded from clinical research studies. It is necessary to include untreated patients with BMs in clinical trials and to evaluate specialized systemic therapy. With the rapid development of lung cancer molecular oncology, an increasing number of key driver genes and corresponding targeted drugs appear, which are expected to further improve the prognosis of patients with NSCLC CNS metastasis. The study of prognostic markers is useful for screening populations with potential therapeutic benefits. These markers can help achieve precise, individualized, and optimized NSCLC CNS metastasis treatment to improve the OS and QOL of patients.

## Figures and Tables

**Figure 1 cells-10-02620-f001:**
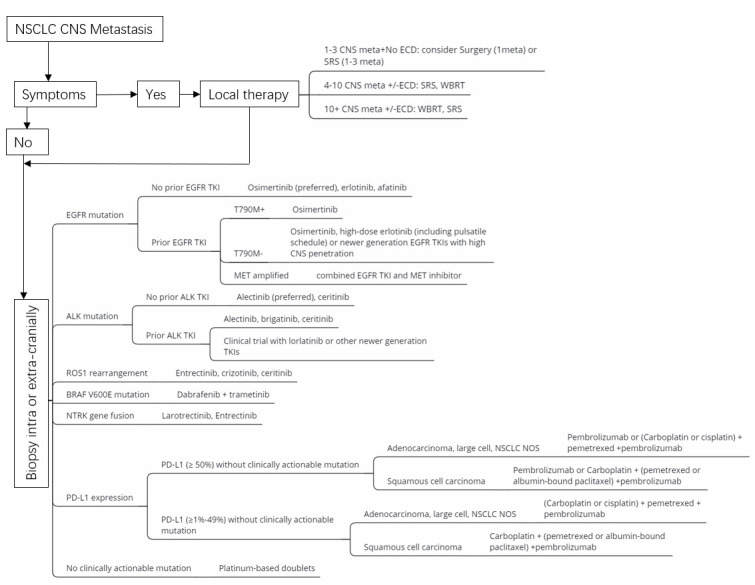
Treatment algorithm for NSCLC CNS metastasis.

**Table 1 cells-10-02620-t001:** Clinical trials evaluating the use of ICIs in NSCLC.

Clinical Trials/NCT Numbers	Drugs	Phase	Sample Size	OS (Months)	Ref
KEYNOTE-001/NCT01295827	Pembrolizumab	I	1260	12	[107]
KEYNOTE-010/NCT01905657	Pembrolizumab	II–III	1034	11.8	[108]
IND-121564/NCT02085070	Pembrolizumab	II	65	7.7	[109]
KEYNOTE-021/NCT02039674	Pembrolizumab	I–II	267	16.7	[110]
	Pembrolizumab combined with carboplatin and paclitaxel			21.4	
	Pembrolizumab combined with carboplatin, paclitaxel, and bevacizumab			16.7	
	Pembrolizumab combined with carboplatin and pemetrexed			16.7	
KEYNOTE-024/NCT02142738	Pembrolizumab	III	305	30	[111]
KEYNOTE-028/NCT02054806	Pembrolizumab	I	477	11.3	[112]
CheckMate-017/NCT01642004	Nivolumab	III	352	9.2	[113]
CheckMate-057/NCT01673867	Nivolumab	III	792	12.2	[114]
CheckMate-063/NCT01721759	Nivolumab	II	140	8.2	[115]
CheckMate-227/NCT02477826	Nivolumab plus ipilimumab	III	2220	17.1	[116]
CheckMate-012/NCT01454102	Nivolumab plus erlotinib	I	472	19.2	[117,118,119]
	Nivolumab			19.4	
FIR/NCT01846416	Atezolizumab	II	138	6.3	[120]
OAK/NCT02008227	Atezolizumab	III	1225	13.8	[121]
POPLAR/NCT01903993	Atezolizumab	III	287	12.6	[122]
IMpower150/NCT02366143	Atezolizumab combined with carboplatin and paclitaxel	III	1202	14.4	[123]
	Atezolizumab combined with bevacizumab, carboplatin, and paclitaxel			19.2	
IMPower-131/NCT02367794	Atezolizumab combined with carboplatin and paclitaxel	III	1021	14	[124]
PACIFIC/NCT02125461	Durvalumab	III	713	28.3	[125]
Study 1108/NCT01693562	Durvalumab	I-II	1022	12.4	[126]
ATLANTIC/NCT02087423	Durvalumab	II	446	13.2	[127]
CAURAL/NCT02454933	Durvalumab and osimertinib	III	29	Not reported	[128]
TATTON/NCT02143466	Durvalumab and osimertinib	I	344	Not reported	[129]
NCT02000947	Durvalumab and tremelimumab	I	459	Not reported	[130]
ARCTIC/NCT02352948	Durvalumab	II	597	11.7	[131]
	Durvalumab and tremelimumab			11.5	
IND226/NCT02537418	Durvalumab	I	153	19.8	[132]

**Table 2 cells-10-02620-t002:** Concentration of tyrosine kinase inhibitors in the cerebrospinal fluid.

Drug Name	Cerebrospinal Fluid Concentration	Cerebrospinal Penetration Rate	Ref
EGFR-targeted therapies			
Erlotinib	28.7 ng/mL (66.9 nM)	2.8–3.3%	[145,146]
Gefitinib	3.7 ng/mL (8.2 nM)	1.13%	[145]
Afatinib	1.4 ng/mL (2.9 nM); 1 nM	1.65%	[147]
Osimertinib	7.51 nM	2.5–16%	[148,149]
AZD3759	25.2 nM	100%	[150]
ALK-targeted therapies			
Crizotinib	0.616 ng/mL (0.14 nM)	0.26%	[84]
Ceritinib	No data	15%	[151,152]
Alectinib	2.69 nM	63–94%	[153,154]
Lorlatinib	2.64–125 ng/mL (6.5–308 nM)	20–96%	[95,152,155]

## Data Availability

Data sharing is not applicable to this article.

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
