# Peer review of "Research Progress and Challenges in the Treatment of Central Nervous System Metastasis of Non-Small Cell Lung Cancer"

_cells, 2021, doi:10.3390/cells10102620_

Round 1

Reviewer 1 Report

 The manuscript “Research progress and challenges in the treatment of central nervous system metastasis of non-small cell lung cancer” by Wang  B et al addresses a relevant clinical question for treatment of advanced NSCLC patients.

The paper is well structured and presented. In order to improve the understanding of the manuscript, the authors should show in the text some representative images derived from published paper in the field, obtained from imaging studies and showing the therapeutic success in metastatic patients. Furthermore, they could add some tables summarizing for each type of anticancer class/drug its efficacy in penetrate the blood-brain barrier and induce a therapeutic response. They should report in the table some clinical parameters indicating the drug efficacy in reducing NSCLC CNS metastasis and improve patient’s survival.

Author Response

Dear Editor,

I revised the original article further. The following is an answer to your comments.

In the revised draft, I further discussed the research progress and challenges in the treatment of central nervous system metastasis of non-small cell lung cancer. The previous manuscript has been embellished by a professional company, and this revised manuscript has been further embellished. I have added 2 tables and 1 figure to the article to make it more readable.

Yours,

Bin Wang

Reviewer 2 Report

Authors provided a review on the management of CNS metastases in NSCLC patients. They divided the manuscript by type of treatment (local treatment, and different types of systemic therapies). This subject is important and actual since the arrival of new targeted therapies and immunotherapy last years for NSCLC.

However, there are major limitations listed above

  • Treatment and management algorithms are missing
  • The optimal timing of the different interventions have to be discussed
  • A table of all studies evaluated ICI on brains metastases in NSCLC should be done.

Introduction:

P1L36: ref 6 is not adapted to the sentence.

P1L42: ref 8-10 are not adapted, authors have to had references regarding ALK or HER2 linked to the associated sentence.

P2L43-46: I think authors have to describe epidemiology and prognosis more precisely regarding type of NSCLC. Prognostic is dramatically different between subtype. P2L45: reference is only for EGFR mutated patients treated with osimertinib, a very specific subtype of NSCLC.

Local treatment:

P2L65 : hormonal treatment ????

One argument in favour of surgery could be the rapid steroid tapering, thus allowing rapid introduction of immune checkpoint inhibitors. It should be discuss.

Post-operative strategies need to be discussed by authors (Mahajan Lancet Oncol 2017, or Brown Lancet Oncol 2017)

P2L74 – ref 18 is not correct.

The paragraph on WBRT is incomplete. Fraction and doses has to be at least cited.

P2L71 : I don’t agree, patients with poor PS should not receive WBRT. Authors should read and cite the QUARTZ trial.

Authors have to precise that there is no WBRT indication for patients with EGFR or ALK mutation.

2.3

Please give more details on doses and fractions of SRS/SRT.

P2L88: Please give numbers

Paragraph 4

P3L129 – ref 33 is not adapted

P3L134: Please add references

P3L137: please add references

P3L142: give numbers compared to non-mutated EGFR patients

P4L146: numbers?

P4L157: references ?

P4L158-160: authors have to describe brain metastases control with osimertinib versus first-line TKI. There is not enough description.

P4L162: which line of treatment?

P4L187 the reference 56 is inappropriate

P5 L231 : it was before the era of ICI

P5L232-252: Authors have to had numbers.

P5L240 : which ones ?

P5L241; do authors have any data regarding anti-HER2 targeted therapies ?

P6L252: first line of treatment or latter line ?

P6L256; first line ? latter line?

Chapter 5.1 talked about ICI monotherapy, but authors presented a retrospective study of ICI in combination with chemotherapy L257

Paragraph on ICI is very important for this topic, and timing with RT. A table of all studies evaluated ICI on brains metastases in NSCLC should be done.

Discussion

P7L331-336 : I’m not sure this sentences should be in the discussion part

Globally, I think that the whole paragraph 6.1 is not really a global discussion, but more a precision regarding patients with oncogenic alterations.

It would be interesting to add ref from the last ESMO guidelines from E. Lerhun, 2021.

Author Response

Dear Reviewer,

I revised the original article further. The following is an answer to your comments.

In the revised draft, I I modified the article according to your suggestion. The previous manuscript has been embellished by a professional company, and this revised manuscript has been further embellished. I have added 2 tables and 1 figure to the article to make it more readable.

  • P1L36: ref 6 is not adapted to the sentence.

Reference 6 has been replaced with “Patil S, Rathnum KK. Management of leptomeningeal metastases in non-small cell lung cancer. Indian J Cancer. 2019;56(Supplement):S1-s9.” based on your suggestion.

  • P1L42: ref 8-10 are not adapted, authors have to had references regarding ALK or HER2 linked to the associated sentence.

Added references “Gainor JF, Ou SH, Logan J, Borges LF, Shaw AT. The central nervous system as a sanctuary site in ALK-positive non-small-cell lung cancer. J Thorac Oncol. 2013 Dec;8(12):1570-3.” and “Koo JS, Kim SH. EGFR and HER-2 status of non-small cell lung cancer brain metastasis and corresponding primary tumor. Neoplasma. 2011;58(1):27-34.” based on your suggestion.

  • P2L43-46: I think authors have to describe epidemiology and prognosis more precisely regarding type of NSCLC. Prognostic is dramatically different between subtype.

I modified the prognostic content based on your suggestion.

  • P2L45: reference is only for EGFR mutated patients treated with osimertinib, a very specific subtype of NSCLC.

Reference has been replaced with “Patil S, Rathnum KK. Management of leptomeningeal metastases in non-small cell lung cancer. Indian J Cancer. 2019 Nov;56(Supplement):S1-S9.”

  • P2L65 : hormonal treatment ????

The hormonal treatment has been replaced with corticosteroid treatment.

  • One argument in favour of surgery could be the rapid steroid tapering, thus allowing rapid introduction of immune checkpoint inhibitors. It should be discuss.

The rapid steroid tapering role of surgery has been described based on your suggestion.

  • Post-operative strategies need to be discussed by authors (Mahajan Lancet Oncol 2017, or Brown Lancet Oncol 2017)

The post- operative strategies have been discussed properly based on your suggestion.

  • P2L74 – ref 18 is not correct.

The ref has been replaced based on your suggestion.

  • The paragraph on WBRT is incomplete. Fraction and doses has to be at least cited.

The Fraction and doses of WBRT has been discussed based on your suggestion.

  • P2L71 : I don’t agree, patients with poor PS should not receive WBRT. Authors should read and cite the QUARTZ trial.

I agreed with you and I’ve changed the sentences based on your suggestion.

  • Authors have to precise that there is no WBRT indication for patients with EGFR or ALK mutation.

I have further emphasized that there is no WBRT indication for patients with EGFR or ALK mutation

  • Please give more details on doses and fractions of SRS/SRT.

The Fraction and doses of SRS/SRT has been discussed based on your suggestion.

  • P2L88: Please give numbers

I’ve given numbers based on your suggestion.

  • P3L129 – ref 33 is not adapted

The ref has been replaced based on your suggestion.

  • P3L134: Please add references

The ref has been added based on your suggestion.

  • P3L137: please add references

The ref has been added based on your suggestion.

  • P3L142: give numbers compared to non-mutated EGFR patients

I’ve given numbers based on your suggestion.

  • P4L146: numbers?

I’ve given numbers based on your suggestion.

  • P4L157: references ?

I’ve given numbers based on your suggestion.

  • P4L158-160: authors have to describe brain metastases control with osimertinib versus first-line TKI. There is not enough description.

I have revised the article accordingly based on your suggestions, and added more description.

  • P4L162: which line of treatment?

It is the first-line treatment in this study.

  • P4L187 the reference 56 is inappropriate

The ref has been replaced based on your suggestion.

  • P5 L231 : it was before the era of ICI

The combination of atezolizumab and bevacizumab with chemotherapy is a therapeutic option for patients with NSCLC CNS metastasis without driver mutations.

  • P5L232-252: Authors have to had numbers.

I’ve given numbers based on your suggestion.

  • P5L240 : which ones ?

The clinical trials have been added based on your suggestion.

  • P5L241; do authors have any data regarding anti-HER2 targeted therapies ?

The data has been added, and HER2 is a novel target for NSCLC.

  • P6L252: first line of treatment or latter line ?

Based on references, ICI is the first-line treatment in this study.

  • P6L256; first line ? latter line?

Based on references, Pembrolizumab is the second-line treatment in this study.

  • Chapter 5.1 talked about ICI monotherapy, but authors presented a retrospective study of ICI in combination with chemotherapy L257

The sentence position has been revised based on your suggestion.

  • Paragraph on ICI is very important for this topic, and timing with RT. A table of all studies evaluated ICI on brains metastases in NSCLC should be done.

The table has been added based on your suggestion.

  • P7L331-336 : I’m not sure this sentences should be in the discussion part

The BBB permeability is of great significance to the treatment of NSCLC CNS metastasis, and it has great influence on the choose of TKIs.

  • It would be interesting to add ref from the last ESMO guidelines from E. Lerhun, 2021.

The ref has been added based on your suggestion.

Thank you very much for your suggestions, I have benefited a lot.

Yours,

Bin Wang